# 'If relevant, yes; if not, no': General practitioner (GP) users and GP perceptions about asking ethnicity questions in Irish general practice: A qualitative analysis using Normalization Process Theory

**Maria Roura[1], Joseph W. LeMaster[2], Ailish Hannigan[3], Anna Papyan[4], Sharon McCarthy[4], Diane Nurse[5], Nazmy Villarroel[6], Anne MacFarlane[3]***

1 School of Public Health, University College Cork, Cork, Ireland, 2 Kansas University Family Medicine Research, University of Kansas Medical Centre, The University of Kansas, Kansas City, Kansas, United States of America, 3 School of Medicine & Health Research Institute, University of Limerick, Limerick, Ireland, 4 Shannon Family Resource Centre, Shannon, Ireland, 5 National Office for Social Inclusion, Health Service Executive, Dublin, Ireland, 6 Department of Sociological Studies, University of Sheffield, Sheffield, England

* anne.macfarlane@ul.ie

## Abstract

### Context

The use of ethnic identifiers in health systems is recommended in several European countries as a means to identify and address heath inequities. There are barriers to implementation that have not been researched.

### Objective

This study examines whether and how ethnicity data can be collected in Irish general practices in a meaningful and acceptable way.

### Methods

Qualitative case study data generation was informed by Normalization Process Theory (NPT) constructs about 'sense' making and 'engagement'. It consisted of individual interviews and focus group discussions based on visual participatory techniques. There were 70 informants, including 62 general practitioner (GP) users of diverse ethnic backgrounds recruited through community organisations and eight GPs identified through an inter-agency steering group. Data were analysed according to principles of thematic analysis using NPT.

### Results

The link between ethnicity and health was often considered relevant because GP users grasped connections with genetic (skin colour, lactose intolerance), geographic (prevalence of disease, early years exposure), behavioural (culture/food) and social determinant

**Data Availability Statement:** All relevant data are within the paper and its Supporting information files.

**Funding:** AMacF HRA-PHR-2015-1344 Health Research Board, Ireland https://www.hrb.ie/ The funders had no role in study design, data collection and analysis, decision to publish, or preparation of the manuscript.

**Competing interests:** The authors have declared that no competing interests exist.

(housing) factors. The link was less clear with religion. There was some scepticism and questions about how the collection of data would benefit GP consultations and concerns regarding confidentiality and the actual uses of these data (e.g. risk of discrimination, social control). For GPs, the main theme discussed was relevance: what added value would it bring to their consultations and was it was their role to collect these data? Their biggest concern was about data protection issues in light of the European Union (EU) General Data Protection Regulation (GDPR). The difficulty in explaining a complex concept such as 'ethnicity' in the limited time available in consultations was also worrying.

## Conclusions

Implementation of an ethnicity identifier in Irish general practices will require a strong rationale that makes sense to GP users, and specific measures to ensure that its benefits outweigh any potential harm. This is in line with both our participants' views and the EU GDPR.

## Background

The health, well-being and health service utilization experiences of some ethnic groups in Europe are poorer than those of others [1–3]. A fundamental step in identifying which specific populations are at greatest need and defining appropriate interventions is the collection and use of reliable data about ethnicity [4,5]. For example, minority ethnic groups have been identified as having higher prevalence rates of diabetes, worse diabetes control, and higher rates of complications than majority ethnic groups. Systematic reviews have shown that culturally adapted interventions can improve the quality of diabetic care for minority ethnic groups and improve health outcomes [6–8].

Biologists have long argued that the objective existence of different human 'races' is not a biological fact or a universal truth because no specific genes can be ascribed to a particular 'race': The use of this concept in population health statistics is hotly contested as a way to perpetuate the false idea that human populations can be clustered in racial groups [1]. Still, 'race' is commonly assigned by society to individuals based on visual characteristics—primarily skin colour but also eye colour and hair texture—and this classification into racial categories has important implications for individuals and societies as a whole [9]. These include well-documented health disadvantages of *racialized* population groups such as Afro-American Blacks, European Roma and Irish Travellers. Irish Travellers are an indigenous ethnic minority group who have poorer health care experiences and health outcomes than the general population [10]. These disadvantages are hard to monitor and address if no data on 'racial' background is collected.

In the European context, ethnicity tends to be a more acceptable concept than 'race' although it is also a slippery concept [11]. Ethnicity has more to do with a shared cultural heritage or ancestry accounting for subjective perceptions about belonging to a particular group. These are not static and may change with time and place. Therefore, the complexity of ethnic identity is difficult to capture in discrete categories because ethnicity is a fluid, flexible, subjective and contextual concept [12]. Against the backdrop of this perennial debate and myriad conceptual, political and practical challenges, the implementation of ethnicity data collection in healthcare systems in Europe is poor [11], with little empirical research on how ethnic identifiers are introduced, embedded and used.

The 2014 Irish Human Rights and Equality Commission Public Sector Duty [13] charges Irish publicly funded bodies with responsibility to conduct equality and human rights assessments and to annually report on evidence of progress in furthering equality. The second Health Service Executive (HSE) National Intercultural Health Strategy [14] and Ireland's Migrant Integration Strategy [15] made recommendations to collect data around ethnicity. A 'no data, no progress' rationale lies behind these recommendations, aimed at identifying which specific populations are most at risk and defining appropriately targeted interventions to reduce health inequities. This is important because, while Ireland has a long history of emigration, it has become an increasingly diverse society since the late 1990s [16]. In addition to the aforementioned indigenous ethnic minority group—Irish Travellers—the last census found that 17% of the total population were born outside Ireland [17]. Migrants are a heterogeneous population. In Ireland, there are EU/European Economic Area (EEA) nationals who are free to move, live and work in Ireland with no special permission; migrants who move for work or study reasons, through marriage, civil partnership or close family relationship; and people seeking international protections and refugees via the Refugee Protection Programme (RPP) established by the Government in September 2015 [18,19]. Hereafter, 'migrant' will be used as a general term unless the specific type of migrant is relevant.

The HSE has proposed a system-level response to record data about ethnicity: an ethnic identifier embedded in existing health information systems [14]. The HSE ethnic identifier is based on the principles of self-identified ethnicity. It utilizes the ethnic and cultural categories, agreed at Government level, employed in the Irish Census of Population. The HSE recommends use of this question as a pragmatic means of assuring consistency and capacity for comparison across data sources. However, both the Central Statistics Office and the HSE have always acknowledged that the concept of ethnicity and its associated recording are very complex and that the census question requires critical reflection and development (HSE Social Inclusion Office, personal communication). The current HSE ethnic identifier also includes questions about country of birth, religion and main language spoken (see Fig 1). These questions are asked as a suite aimed at forming a holistic approach to identifying and addressing needs of patients from diverse groups. To support implementation of the ethnic identifier, the

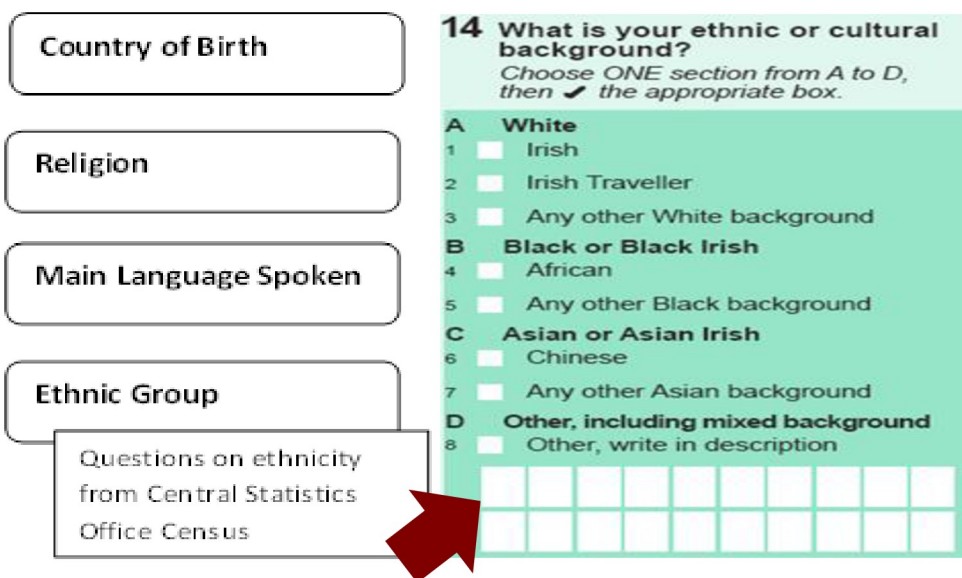

**Fig 1. Irish Health Service Executive ethnic identifier.**

HSE Social Inclusion Office has prepared training materials and information leaflets, translated into several languages, and proposed pilot implementation projects.

Despite this, ethnicity data are not routinely collected in Ireland. Ethnic identity is recorded in only 14 of 97 (14%) national health and social care data collections. Ethnicity is collected in psychiatric hospitals, drug and alcohol treatment centres and for some infectious disease notifications [20], but there is no routine data collection in primary care settings. While there is an increase in Irish research about migrant health, there has been little research on ethnicity data collection [18]. As part of a larger study [21], this paper focuses on levers and barriers to implementation of an ethnic identifier in Irish primary care. Informed by an internationally recognized theory of implementation, May and Finch's Normalization Process Theory (NPT: described further below) [22], we present the views of general practice users and general practitioners (GPs) about the routine collection of ethnicity data in Irish general practices. We draw on in-depth analysis of three cases that are used as a *vehicle* to learn about the collection of ethnicity data in primary care in Ireland. The examination of these three specific cases can serve to draw broader lessons that can be transferred to other general practice and primary care settings (*conceptual* generalizability).

## Methods

The Irish College of General Practitioners Research Ethics Committee approved this study and written consent was obtained from participants.

### Methodological approach

This paper is based on a participatory health research study, planned and governed by an inter-agency steering group comprising non-governmental organisation (NGO), academic and health sector partners [21]. Conceived as a theoretically informed case study, it aims at understanding a particular case as an integrated whole in its usual day-to-day context [23]. This is particularly useful when rich, in-depth and highly contextualized accounts of *what is going on* within everyday settings are needed. Where many variables need to be accounted for simultaneously and the boundaries between phenomenon and context are blurred, a *real-life* case study design is likely to yield more useful information than an experiment where the researchers control the context [24]. We envisaged that a case study design was the most appropriate to provide a detailed and nuanced account of the subjective, dynamic, multifaceted, contextual and intertwined nature of ethnicity, as well as a multi-perspectival account of how the introduction of an ethnic identifier is implemented and received in the everyday context of Irish general practice. Therefore, we adopted a relativist perspective to examine the implementation of collecting data about ethnicity by considering the complex links between its different dimensions as represented by both *insider* (emic) and *outsider* (etic) views. Similarly, because experiences during provider—user encounters at health care sites are jointly constructed through mutual interaction, we adopted a social constructionist epistemological perspective.

### Theoretical framework

NPT [22] describes four constructs to explain the *work* that stakeholders have to do, individually and collectively, to embed a new way of working in their routine day-to-day practice (Table 1).

We employed NPT's constructs about sense-making and engagement as an overarching framework to inform the study design, data generation and analysis. These two constructs relate to the work that stakeholders have to do to 'think through' what a proposed new way of

**Table 1. NPT constructs to explain what promotes or inhibits implementation.**

| SENSE-MAKING | *Does it make sense to stakeholders?* |
|---|---|
| ENGAGEMENT | *Will stakeholders buy into it and drive its implementation forward?* |
| ENACTMENT | *Do stakeholders have the resources and capacity to put it into practice?* |
| APPRAISAL | *Can stakeholders determine its impact and identify any benefits?* |

working would mean for them [25]. This helped us to keep our study *focused* on important forms of implementation work while still open to *emerging unexpected issues*.

## Sampling and data collection

We identified three *selected cases* that offered a comparison of primary care settings in city, town and rural areas with combined representation of Irish Travellers and migrants. We followed the principles of *purposeful sampling* to select relevant cases to allow us to answer our research question. A *maximum variation criterion* was applied to get maximum variation on dimensions of interest [24]. By studying three cases simultaneously within their respective socio-economic and demographic contexts, we attained a broad understanding beyond an in-depth description of each particular case (see Table 2).

We employed a combination of purposive, convenience and snowball approaches to recruit GP users and GPs at the selected sites. We circulated information about our study through our networks of community groups near the sites. We used academic and community networks to approach GP practices. We organized interactive recruitment sessions to describe the research and explain the informed consent process between November 2017 and June 2018.

We conducted 10 focus groups with 62 GP users between March 2018 and July 2019. An interpreter was used in focus groups with Farsi, Russian and Arabic speaking participants. All other data generation was through English.

**Table 2. Description of case study sites.**

| General Practices | Socio-demographic Data^ | Diverse and Minority Ethnicities* | Diverse Migrant Typologies |
|---|---|---|---|
| Site A | • In a city with the country's most disadvantaged urban areas<br>• Population size of city: 94,192<br>• Ethnicity of electoral division of the city where practice is based: 49% White Irish, 26% any other White background, 9% Asian, 2% Black. | Many, including black Francophone from Cameroon and the Democratic Republic of Congo, and black Anglophone from Nigeria. | Asylum seekers living in State's Direct Provision accommodation, refugees, economic migrants. |
| Site B | • In a town in a region categorized as affluent<br>• Population of town: 9,729<br>• Ethnicity of town where practice is based: 74% White Irish, 14% any other White background, 5% Asian. | Very diverse population, including about 40 nationalities. Not many African. Some Irish Travellers. | Recently arrived war refugees from Syria, Chilean political refugees from the 1980s, economic migrants. |
| Site C | • In a village in a rural area<br>• Population of village: 1,045<br>• Ethnicity of electoral division where the practice is based: 78% White Irish, 10% any other White background, 5% Irish Traveller. | Many from Western Europe, e.g. France, Germany. Also many Polish and Moroccan. Some Irish Travellers. | Syrian refugees |

^Source: CSO Census of Population, 2016; Ethnicity for the population: 82% White Irish, 10% any other White background, 2% Asian, 1% Black, 0.7% Irish Travellers.
*Source: GP descriptions of their practice populations.

At the beginning of the focus groups, we explained to participants that the HSE was interested in ethnicity data collection as an equality monitoring measure to examine health outcomes and inform service adaptations in Ireland. We showed participants the ethnic identifier (see Fig 1).

We employed visual, interactive participatory techniques during the focus groups and a respectful mode of engagement [26] to elicit their views about potential benefits and challenges relating to the collection of ethnicity dataThe techniques we used were based on card sorts and rating exercises [27] and explored sense-making and engagement. The card sort explored *'what are your opinions about the ethnicity questions proposed by the HSE?'* Participants were asked to explain their views to the group and to indicate on a banner whether they were positive (green), unsure (orange) or negative (red) about each component of the HSE ethnic identifier. This 'traffic light' technique was important for eliciting visual details of sense-making and engagement (see S1 File for photo of a sample completed card sort traffic light exercise). The rating exercise was designed to get an indication of overall favourability of the implementation of an ethnic identifier: in a scale of 0 to 10, where 0 is totally against and 10 totally in favour, *where do you stand regarding the collection of ethnic identifiers in primary care?* (See S2 File for photo of a sample completed rating exercise). The related fieldwork guidelines are available as S3 File.

Opportunities for group data collection with GPs was limited due to their busy clinics. There were eight GP participants in total. We conducted four one-to-one semi-structured interviews and four informal meetings with four GPs (one GP at Site A; one GP at B and two GPs at site C). We completed a focus group at one site (Site A) with four GPs. The semi-structured interviews and focus groups were based on a topic guide designed to gather in-depth information about the practice setting and to explore sense-making and engagement (See S4 File). These were audio-recorded. The informal interviews were short meetings with GPs about the research and its progress. These were recorded as researcher debriefing notes.

The complete dataset for analysis included interview transcripts, focus group transcripts, researcher debriefing notes from recruitment and fieldwork sessions, researcher field notes from meetings with GPs, and over 300 brief-notes from PLA fieldwork charts.

The first author (MR), who conducted all the interviews and facilitated focus groups, imported fieldwork data into the qualitative software NVIVO-10 and reviewed all the transcripts following the principles of thematic analysis [28] adopting a mix of inductive and deductive approaches to develop a coding framework that was informed by NPT. This process was supported by regular data analysis meetings between the first and last authors.

The analysis led to the identification of four key emergent themes: *rationale, use, measurement and practicalities*, which were systematically applied to the data set. Findings were presented to the Steering Group members (co-authors) for discussion and interpretation. Results are described in detail in the next section and we discuss the resonance of the themes with NPT in the Discussion section.

## Findings

There were 70 participants in total: 62 GP users of diverse ethnic backgrounds (see Table 3) and eight GP participants (four females and four males aged 30–60 years). GP participants were all white Irish. Table 4 provides details of the GP users' self-identified ethnicity.

Below, we first present findings for each of the emergent themes from GP users followed by GPs. From greatest to lesser themes (i.e. how much the theme was discussed), these were rationale, use, measurement and practicalities for GP users and rationale, practicalities, use and measurement for GPs.

**Table 3. Description of GP users.**

| Site | Group description | No. and gender (F/M) (N = 62; 47F/15M) | Age Range (years) |
|---|---|---|---|
| A | 1. Afghan | 2 (2F) | 35–50 |
| | 2. Mixed | 10 (6F 4M) | 25–65 |
| | 3. Irish Traveller | 4 (4F) | 40–65 |
| | 4. African | 18 (12F 6M) | 25–60 |
| B | 1. Mixed | 4 (4F) | 30–50 |
| | 2. Syrian | 3 (3F) | 25–35 |
| | 3. Irish Traveller | 7 (5F 2M) | 20–60 |
| | 4. Mixed | 4 (3F 1M) | 19–65 |
| | 5. Polish | 4 (3F 1M) | 35–45 |
| C | 1. Mixed | 6 (5F 1M) | 40–65 |

## GP users: *Rationale*

The link between ethnicity and health was often considered relevant because participants grasped connections with genetic (skin colour, lactose intolerance), geographic (prevalence of disease, early years exposure), behavioural (culture/food) and social determinants (housing) grounds.

> *Very good [to ask about ethnicity], because of diverse culture and diversity in doing things to meet our cultural needs*
>
> *(Site A, FG4 Black African group)*

> *Well I'm thinking about what you said about if it is related to certain diseases coming from certain part of the world. Yeah, that's a good point*
>
> *(Site A, FG1 Afghan group)*

> *If you do go to a doctor or a hospital that they do ask you are you an Irish Traveller or what your background is because of for instance a lot of the Irish Travelling children when they are*

**Table 4. GP user participants' self-identified ethnicity.**

| Ethnicity Category from HSE Ethnic Identifier | Number (%) |
|---|---|
| A **White** any other white | 17 (31.4) |
| B **Black or Black Irish** African | 13 (24)) |
| A **White** Irish Traveller | 10 (18.5) |
| C **Asian or Asian Irish** any other background | 7 (12.9) |
| D Other including mixed background^ | 3 (5.5) |
| B **Black or Black Irish** Black other | 2 (3.7) |
| A **White** Irish | 1(1.8)^^ |
| C **Asian or Asian Irish** Chinese | 1 (1.8) |

^1 Latin American; 2 Other.

^^ Syrian participant who identified as Irish.

*born are put on soya based milk in the hospital. They're not going down with the normal milk because they have lactose intolerance*

*(Site B, FG2 Irish Traveller group)*

*Even if the child is born in Ireland but the parents would still do their own cultural things. Like you know a lot of African parents . . . That's the way they do things at home and even if the kids are born here in Ireland they would still do*

*(Site B, FG 1 Mixed group)*

There were mixed findings about the relevance of asking about religion, which was often contested as irrelevant data in the context of primary care.

*I just don't know what is the relationship between religion and health? I can't understand that part*

*(Site A, FG 2 Mixed group)*

*A doctor's going to be trained in the particular health background needs of every single religion here? Because why would you be collecting that unless you're confident that the GP will be able to say 'this one's an atheist, okay that means da, da, dah'. This one's a Hindu*

*(Site C, FG1, Mixed group).*

There was a strong negative view in the Black African group about country of origin as a question. They explained that they had negative experiences in healthcare settings when their country of origin was known because of associations, for example, between Sierra Leone and Ebola virus outbreaks. They also connected the question to discrimination and 'othering', particularly for their children who had been born in Ireland and faced questions about their origin.

*And I keep saying that I am from Ireland. The person says 'no, where are you originally from?'. You are already creating like you're confusing the minds of the children*

*(Site A, FG4 Black African)*

Although many participants did not strongly oppose the collection of ethnicity data, the rationale of 'improving service provision' mentioned in the HSE information leaflets was not felt to be a convincing argument. They questioned the value of the specific questions being asked and offered examples of other questions that would more obviously lead to information that could be used to improve service provision in primary care (see Table 5).

**Table 5. Main critiques and proposed alternatives to the HSE ethnic identifier questions.**

| Question in HSE Ethnic Identifier | Alternative Question Proposed by GP Users | Rationale for the New Question |
|---|---|---|
| What language you speak at home? | Do you need an interpreter? | You can speak one language at home and also speak English |
| What is your country of birth? | Where have you been the last year? | You may not have been in your country of birth for years but instead visited a tropical country posing specific epidemiological risks (e.g. malaria) |
| What is your religion? | Do you have any food requirement? Do you have any preference over the sex of your gynaecologist? | What has it to do with health? Why not ask food requirements or preference over sex of gynaecologist (not only for Muslims but to all population) |

## GP users: *Use*

There was some scepticism across groups about how the collection of ethnicity data would benefit users. They had concerns regarding confidentiality. They were not convinced that these data would be actually used at all. They reflected that there are inequalities and problems that are known about but are still not addressed. Examples included problems with the provision of interpreters in healthcare settings and addressing Irish Travellers' housing needs. There were also concerns about the potential misuse of ethnicity data. Would it ever be used against communities? The importance of explaining clearly *why* these data needed to be collected and what it would be *used for* emerged as the most critical point.

> *You don't want to be walking into a booby trap [by completing an ethnic identifier in healthcare]*
>
> *(Site A, FG4, Black African group)*

> *Sometimes this data could be stolen and used somewhere else*
>
> *(Site B, FG4 mixed group)*

> *So if you're going into a GP and he asks or she asks these questions you don't mind maybe giving those answers to a GP. But if the information is on a database that is then accessible by the police . . . I think the big question is why do they want the information because I'm not convinced it comes to better service*
>
> *(Site C, FG1, Mixed group)*

## GP users: *Measurement*

Regarding the ethnic categories used in the HSE ethnic identifier, many participants had no concerns. Some participants did raise concerns that the categories were too focused on distinguishing between Irish and non-Irish, leading to 'othering'.

> *The thin[g] is, you [have] the fact that you have to write Black or Black Irish, I think something is wrong [others agree]. So the fact that you Black Irish, Polish Irish, something is wrong like*
>
> *(Site A, FG4 Black African group)*

> *I mean I think of the little girl I know, she's not little any more she's all grown up, her daddy was Hawaiian and mother was Irish. What's she going to put? Other! [. . .] she's Irish [. . .] So the categories are strange. I can't figure out why Chinese are separated unless Chinese have particular health issues that you want to know about*
>
> *(Site C, FG1 Mixed group)*

They reflected that the questions were 'strange' because they mentioned mixed colour and cultural features, were inadequate for reflecting mixed ethnicities and could lead to labelling particular populations.

> *Could the doctors ask a question like 'do you know of any illnesses in the family?' without asking 'are you a Traveller?', without asking 'are you Muslim?', without asking 'what religion you*

*come from?' Can they just ask do you know if there is a specific problem in your family and you can then elaborate and say yes or no and tell them the problem. Instead of just assuming because you're a Traveller that there's a chance you're going to have a bad heart, you're going to have cancer, you're going to have diabetes? It's prevalent to the person, not the group*

*(Site B, FG2 Irish Travellers group)*

## GP users: *Practicalities*

There were no marked preferences on the part of patients about being asked by GPs or other medical staff; however, asking 'nicely' and providing a non-compromising way to decline to answer was very important to participants.

*Ask nicely, treat people with respect, that's all*

*(Site A, FG3 Irish Traveller group)*

## GPs: *Rationale*

None of the GP participants were routinely collecting ethnicity data at the time of the study. Some explained that they had not thought much about it previously. They were interested in the idea and open to exploring how they could pilot it in their day-to-day work by, for example, modifying their information systems to prompt them to gather these data. They were, however, not convinced about the relevance of the questions *for their consultations*. They explained that they gather this information in consultations when it is clinically necessary and, further, that they feel they 'know' their patients. They explained that asking for these sensitive details could disrupt relationships and they had questions about whether it was their role to collect these sensitive data for population health reasons.

*I can't think of any reason why not to do it if people have the option of declining to answer . . . I hadn't thought about it until we met and so I have to say that I suppose we have a huge amount of paperwork to do but considering it does have an impact on the clinical needs of the patient then, yes, it's probably something that we should be collecting.*

*(Site A, GP1; interview)*

*I mean you can say it's [ethnic data collection] positive [. . .] But on a day-to-day basis I would see it as being largely pointless because I don't think it's necessary because we know our patients [. . .] When I come in I know my lady is from Kurdistan, I understand Kurdistan and I know what it is, I know where it is and I know what they've come from. If someone's a Traveller, I know they're a Traveller, I understand what it is to be a Traveller.*

*(Site A, GP1: interview)*

GPs questioned the added value of collecting these data without simultaneously collecting data about the life conditions of people (the social determinants of health) and linking them to the (confidential) data contained in medical histories.

*How can you categorize people in such a way that you'll get the maximum, it will mean you'll get the maximum information out of it? Because you might have to say if you are talking to*

*Travellers you might want to know if they are housed or not housed [. . .] That's because these different social conditions will make a difference*

*(Site C, GP1 interview)*

## GPs: *Practicalities*

The difficulty in explaining a complex concept such as 'ethnicity' in the limited time available in consultations emerged as an important concern. GPs proposed alternative ways of collecting these data, such as using data already collected in the census or conducting qualitative research. Others did not see time as an insurmountable obstacle but emphasised that collection of these data always had to be subject to the users' consent, which was challenging to obtain from non-English speakers or patients with low literacy.

*I see the merit in that [collecting ethnicity data] but it has to be explainable in 30 seconds by the administrator at the front desk because they'll be the ones that will be asked . . . when they're giving out the form and say religion is here and somebody says 'why do you want to know my religion?' there needs to be a short answer*

*(Site A GP focus group)*

*I have patients from the Travelling community who have literacy problems they have to bring their forms to me. I help them fill them out sometimes. They're not able to fill out the forms or read the questions. And then I have patients who have no English so if your form [for collecting ethnicity data] isn't available in their language of choice they're not going to be able to fill that out either, they won't understand the question so it won't be possible*

*(Site B, GP1 interview)*

*Why don't they just take it from the census? I just wonder if you'd get more actual practical information out of a focus group of people that are dealing with the different ethnic groups all the time [. . .] and you could do that in 90 nationalities and you would get much more information from that, lots of individual things*

*(Site A GP focus group)*

## GPs: *Use*

GPs had concerns about data protection, particularly in light of EU GDPR, which was coming into effect shortly after the fieldwork.

**GP4.** Yeah but you don't access their [patients'] records without getting their permission.

**I1.** Yeah. And this is one of the topics that we should decide how we are going to do with this data protection, but initially the idea . . ..

**GP4.** But that's straightforward they have to give permission for you to access their records. There's no discussion about that, that's reality. And even more so with this new data protection thing coming in NEXT month (Site A, GP focus group).

Conversely, one GP considered that the new legislation provided an opportunity to improve data protection, including in this area:

*Well I know for all new patients because of GDP law, that new EU law that came in on the 25th May [2019]. We actually have a template, it's a new patient registration form that*

*includes data protection issues so it's something that we could possibly, we would probably need to include that on it. Can we collect and hold data on your ethnicity and language and'/ or religion?*

*(Site B, GP1 interview)*

There were also concerns among GPs that these data might be used to blame GPs for inequalities that are intrinsic to the two-tier Irish health care system and broader socio-economic context.

*The inequities come in the HSE side when you're trying to access services and not necessarily who they are or where they're from it's often like just because it's the natural dual, two tiered health system that we have*

*(Site A GP1 interview)*

### GPs: *Measurement*

There were some concerns about the validity of the categories used, and the fact that GP practice management systems were not readily able to incorporate data about ethnicity according to the census categories.

*What do you mean by ethnicity? You know this is a really [. . .] if you have somebody who is Black, if you write it down ethnicity is Black, well what does Black mean? Does it mean African Black, does it mean Indian Black, does it mean Caribbean, Afro American or does it mean mixed race?*

*(Site C, GP1 Interview).*

At the end of each focus group with GP users and interviews with GPs, participants were asked to reflect on their overall view of an ethnic identifier in Irish primary care, and to rate it on a scale of *0–10; 0 means totally against and 10 means totally in favour*. Three GP interviewees gave ratings ranging from 6 to 10. Ratings by 51 GP users ranged widely, from 0 to 10 with a median of 6.

## Discussion

Our study found that there are a multiplicity of views about implementation of an ethnic identifier in Irish general practice settings across GP users and among GP participants. Notwithstanding some differences about specific issues, we found that that these stakeholders can see the potential value of collecting ethnicity data at the population level. They have clear concerns, however, about the relevance and importance of doing so *in general practice settings*. They are also concerned that these data collected will not be used or that it will be misused. The measurement categories for ethnicity are contested as well. There are specific concerns among GPs about the practicality and ease of asking these questions in their time-constrained practices as well as data protection issues. When asked to rate the favourability of implementing an ethnic identifier, there was considerable variation within and across groups.

Previous research has found that some minority ethnic groups, particularly those with experience of being persecuted, can be very opposed to ethnicity data collection [11]. In contrast, Ireland's national organization for advocacy for Irish Travellers is strongly in favour of this for

equality monitoring purposes [29]. Many participants from diverse ethnic backgrounds in this study were also broadly in agreement that there is potential value in having these data at a population level. The concerns of those who were less favourable, however, are notable. The finding that there were different concerns about different questions in the ethnic identifier is notable too. For example, the question about religion was considered particularly contentious by some; the question about country of origin was particularly problematic for Black African participants. These findings highlight how important it is for people using GP care to know that they are not obliged to complete some/all of the questions in the HSE ethnic identifier and that their healthcare and welfare otherwise will not be compromised if they do. It is critical to think about how these messages can be conveyed adequately to people with low literacy and limited English. This is in the context of a health care system with diminishing resources for Traveller health [30] and no routine use of trained interpreters [31].

Fears among GP users and GPs about data being collected and not being used appropriately have been documented in other countries [32]. There is evidence of a problematic pattern of data under-use in Ireland [20]. One of the principles of GDPR, data minimization, requires that only data that are needed are collected, so collecting data and then not using them undermines this principle. Further, fears about misuse were also recorded in this study. They resonate with similar international research about the harms and benefits of ethnicity data collection in healthcare settings [33]. These fears are understandable in the current political climate where health data were used for immigration control purposes in the UK [34]. The collection of ethnicity data requires trust in the stated reason for its collection. This entails comprehensively informing patients of all potential uses of these data, including the potential of sharing data with public bodies for infectious diseases such as COVID-19. Trust that data will not be used for purposes for which they were not collected is also important [11]. Again, the implications of obtaining informed consent without adequate supports for patients with literacy and language issues is particularly challenging.

In keeping with the literature [12], our findings show that the measurement of ethnicity is complex and contested. Despite the complexity of ethnicity as a concept and the associated measurement challenges, Farkas concludes that if categories are carefully applied and interpreted they can reflect that complexity [11]. Moreover, where ethnicity data are collected and used there is potential for positive developments, e.g. increased awareness of the impact of persistent ethnic differences and interventions to improve preventive care and access to healthcare to address these issues [35]. In Ireland, secondary data analysis of the National Drug Treatment Reporting System, which routinely collects information on ethnicity, provided evidence on addiction services accessed by Irish Travellers. This was then used to inform a training intervention for staff and peer workers to improve their access to services [36,37].

Findings from GPs about time pressures and the lack of resources in their setting as a barrier to implementation work for an ethnic identifier resonate with the UK setting [38]: incentivizing GPs was a significant lever to their engagement with this work [5]. We know from the literature that there are problems in securing resources in primary care for implementation of interventions relating to migrant health specifically (e.g. providing interpretation services [39]), as well as the general population [40].

Taken together and following NPT, this analysis indicates that there are major challenges to be addressed before the collection of ethnicity data will be integrated into routine day-to-day GP practice in Ireland. Although they recognized its potential value, many GP users and GP participants questioned the rationale for collecting these data (low sense-making) and GPs did not 'buy into' the idea that it was their role to collect it (low engagement). In addition, findings that emerged about enactment (see Table 1) are noteworthy. Thinking through the practical aspect of ethnicity data collection in daily practice (time pressures, lack of support for working

with low literacy and language barriers, need for adapting health information systems, concerns about patient interactions and relationship building), there are serious obstacles. Given the importance of sense-making and engagement to initiate implementation work [22,41] and the need for supportive contexts to resource implementation work [42], these findings do not bode well for the successful implementation of an ethnic identifier in Irish general practice at this time.

## Strengths and limitations

Efforts were made to recruit GP users from a variety of ethnic backgrounds and to recruit GPs at all three selected sites. In keeping with best practice for qualitative research, the approach to sampling was robust in terms of using combined approaches (maximum variation sampling for site selection, purposive and snowball sampling for participants) to recruit 'information-rich' 'participants and to reach data saturation. There are more females than males in the sample and we cannot know if a sample with more male participants would have produced different results.

The focus groups were facilitated by an experienced researcher who enabled participants to share perspectives. Co-facilitation by community partners from NGOs who had established relationships with some groups was also a strength. As indicated throughout the results, participants reported and discussed a variety of perspectives and did reflect on each other's views.

The involvement of academic, community and health sector partners in data interpretation, in line with our participatory health research approach, strengthens the trustworthiness of findings. Our use of NPT as a heuristic device to 'think through' data generation and analysis, using a conceptual framework, is also a strength. It ensures comprehensive analysis of implementation work and adds to the conceptual generalizability of the research.

## Implications for policy and practice

These findings could be used to design interventions to improve perceived understanding of the rationale for an ethnic identifier among ethnic minority groups and GPs. Given that participants could not necessarily see the link between the ethnic identifier questions and service improvement (Table 5), findings could be used to appraise and adapt HSE resources (information leaflets and training materials) to strengthen this link further. Following Farkas [11] and Pavee Point [29], we recommend collaboration between the HSE, the national professional bodies for GPs and ethnic and migrant community organisations to see if it is possible to provide a coherent message about the value of ethnicity data collection in general practice settings. Further, the findings of scepticism about the rationale of monitoring to address health inequities must be considered. Making the case for ethnicity data collection to support the reduction of inequities will also require political commitment to act on the available evidence [35].

The ethnic identifier itself will need updating as the measurement of ethnicity in the Census of Population changes. The current question, used since 2006, will change for the next Census following a consultation process with members of the public, advocacy groups and government organisations [43].

Regarding data protection, the potential for non-use/misuse of the data may vary depending on the broader political climate in Ireland as well as other EU countries where there is reluctance to collect this type of data [11]. The 2014 Public Sector Duty [13] is, however, an important legal instrument to underpin best practice, implementation and use of ethnicity data collection to monitor discrimination against ethnic minorities in the Irish health care system. Again, intersectoral partnerships and action are likely to be important to advocate for the

appropriate collection and use of ethnicity information as per anti-discrimination laws and instruments [44].

Linked to this and looking at the specifics of Ireland and the impact of GDPR legislation introduced during the course of this study, there are significant additional problems not captured in the fieldwork. While exploring the implications of participants' concerns about GDPR for our research, we learned that the Data Protection Working Group for the largest professional association of GPs viewed ethnicity data as highly sensitive and did not consider that GPs had a role in routine collection of ethnicity data (Irish College of General Practitioners Information Technology Group, personal communication, February 2019). This is now an additional barrier to the HSE's plan to implement an ethnic identifier in primary care. There needs to be considerable collaboration and negotiation between the HSE, the GP professional bodies, GPs and relevant community organisations to clarify this issue and ensure that ethnicity data can be collected safely and confidentially. If this is achieved, GDPR may support implementation in other ways because it raises standards about data collection, storage and sharing due to the penalties in place.

## Conclusion

Intersectoral strategies to improve perceived rationale and to ensure appropriate use of ethnicity data are necessary to strengthen sense-making and 'buy in' for implementation of an ethnic identifier in Irish general practice. Further, as the impact of GDPR legislation has shown, the context for implementation is very fluid and requires ongoing critical analysis.

## Supporting information

**S1 File. Photo PLA traffic light exercise.**
(TIF)

**S2 File. Photo PLA rating scale.**
(TIF)

**S3 File. Guideline PLA community fieldwork.**
(TIF)

**S4 File. Guideline interview GP fieldwork.**
(TIF)

## Acknowledgments

We are grateful to all community gatekeepers and research participants for their time and interest. We thank EMH-IC Steering Group members Colette Bradley, Aisling Romer, Alphonse Basogomba and Maura Adshead for support with the research.

## Author Contributions

**Conceptualization:** Joseph W. LeMaster, Ailish Hannigan, Diane Nurse, Anne MacFarlane.

**Formal analysis:** Maria Roura.

**Funding acquisition:** Joseph W. LeMaster, Ailish Hannigan, Diane Nurse, Anne MacFarlane.

**Investigation:** Maria Roura, Anna Papyan, Sharon McCarthy, Anne MacFarlane.

**Methodology:** Maria Roura, Anne MacFarlane.

**Project administration:** Maria Roura, Anne MacFarlane.

**Resources:** Maria Roura.

**Supervision:** Maria Roura, Anne MacFarlane.

**Writing – original draft:** Maria Roura.

**Writing – review & editing:** Maria Roura, Joseph W. LeMaster, Ailish Hannigan, Anna Papyan, Sharon McCarthy, Diane Nurse, Nazmy Villarroel, Anne MacFarlane.

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
