## [Decision Letter · Decision Letter 0]

24 Feb 2021

PONE-D-20-17815

‘If relevant, yes; if not, no’: General practitioner (GP) users and GP perceptions about asking ethnicity questions in Irish general practice: A qualitative analysis using Normalization Process Theory

PLOS ONE

Dear Dr. MacFarlane,

Thank you for submitting your manuscript to PLOS ONE. After careful consideration, we feel that it has merit but does not fully meet PLOS ONE’s publication criteria as it currently stands. Therefore, we invite you to submit a revised version of the manuscript that addresses the points raised during the review process.

We look forward to receiving your revised manuscript.

Kind regards,

Barbara Schouten

Academic Editor

PLOS ONE

Journal Requirements:

Reviewers' comments:

Reviewer's Responses to Questions

**Comments to the Author**

1. Is the manuscript technically sound, and do the data support the conclusions?

Reviewer #1: Yes

Reviewer #2: Yes

2. Has the statistical analysis been performed appropriately and rigorously? 

Reviewer #1: N/A

Reviewer #2: N/A

3. Have the authors made all data underlying the findings in their manuscript fully available?

Reviewer #1: Yes

Reviewer #2: Yes

4. Is the manuscript presented in an intelligible fashion and written in standard English?

Reviewer #1: Yes

Reviewer #2: Yes

5. Review Comments to the Author

Reviewer #1: Overall, this is a well-written, clear and well-argued article that merits publication. I had very few comments, namely:

- In the introduction, following paragraph 2: are the Irish Travellers’ normally considered a different ‘racial’ group in Ireland or is it more likely that they are a different ethnic group? It strikes me that they would probably still be considered Caucasian in research? (Just to clarify: in my experience, people tend to refer the term ‘race’ to the large continental groups – Caucasian, African, Asian etc – and the term ethnicity to other differences within those very large groups);

- This is a very minor comment, but there were a few misplaced commas and a missing full stop on pg 11. Otherwise I didn’t notice any missing punctuation or typos;

- Pg 11 3rd paragraph: it would be good if you could add a couple of sentence about a) whether participants were familiar with the HSE guidance and proposed categories and b) whether they knew of the potential benefits and challenges relating to the collection of ethnicity data. I imagine that was a precursor to the FGDs but it would be good if you could clarify this. Providing such information is particularly important because the quotes in the article reveal that people may not have fully understood the reason for asking questions about ethnicity (see also my comment on Table 5) and because of the sensitivity of the questions during the interviews;

- What is the difference between ‘IDIs with GPs’ and ‘informal meetings with GPs’?

- The proposed alternative questions if Table 5 are quite interesting, because at least to me they suggest that the participants may not have understood what the HSE questions are trying to get at. The reframed questions seem to be directly either about the service experience (e.g. do you prefer a male or female gynaecologist; do you need an interpreter) or about things that more directly link to health (what foods do you eat; is there something in your travel history that could explain your symptoms) – but obviously, that is not what the HSE questions are trying to get at. If you decide to keep Table 5 and the immediately preceding paragraph, then I think that you do need to comment on this in the main text of the article, and offer a bit of a reflection about what this means for your overall results? Is this a limitation to your study perhaps?

Reviewer #2: A study that is useful in Primary Care beyond the Irish context. Could have been even better if paired with a systematic review of reasons (see Sofaer and Strech) as for example used by Moscrop et al to look at socio-economic (asking and coding) questions in the consultation. Whilst the number of GPs participating was low for this kind of study, the reasons for this make sense.

6. PLOS authors have the option to publish the peer review history of their article (what does this mean?). If published, this will include your full peer review and any attached files.

Reviewer #1: **Yes: **Jantina De Vries

Reviewer #2: No

---

## [Author Response · Author response to Decision Letter 0]

19 Apr 2021

PONE-D-20-17815

‘If relevant, yes; if not, no’: General practitioner (GP) users and GP perceptions about asking ethnicity questions in Irish general practice: A qualitative analysis using Normalization Process Theory

PLOS ONE

Response to reviewers

April 2 2021

Reviewer #1: Overall, this is a well-written, clear and well-argued article that merits publication. 

RESPONSE: Thank you for this positive feedback.

- Reviewer #1: In the introduction, following paragraph 2: are the Irish Travellers’ normally considered a different ‘racial’ group in Ireland or is it more likely that they are a different ethnic group? It strikes me that they would probably still be considered Caucasian in research? (Just to clarify: in my experience, people tend to refer the term ‘race’ to the large continental groups – Caucasian, African, Asian etc – and the term ethnicity to other differences within those very large groups);

RESPONSE: In the Irish context, Irish Travellers are defined as a minority ethnic group – see reference 10. 

- Reviewer #1: This is a very minor comment, but there were a few misplaced commas and a missing full stop on pg 11. Otherwise I didn’t notice any missing punctuation or typos;

RESPONSE: We have edited some commas and added the missing full stop on page 11. 

Reviewer #1: Pg 11 3rd paragraph: it would be good if you could add a couple of sentence about a) whether participants were familiar with the HSE guidance and proposed categories and b) whether they knew of the potential benefits and challenges relating to the collection of ethnicity data. I imagine that was a precursor to the FGDs but it would be good if you could clarify this. Providing such information is particularly important because the quotes in the article reveal that people may not have fully understood the reason for asking questions about ethnicity (see also my comment on Table 5) and because of the sensitivity of the questions during the interviews; 

RESPONSE: At the beginning of the focus groups, we explained to participants that the HSE was interested in ethnicity data collection as an equality monitoring measure to examine health outcomes and inform service adaptations in Ireland. We showed participants the ethnic identifier (see Figure 1). 

The focus groups were designed to elicit their views about the potential benefits and challenges.

This information has been added to the paper, page 11. 

Reviewer #1: - What is the difference between ‘IDIs with GPs’ and ‘informal meetings with GPs’? 

RESPONSE: The semi-structured interviews (and focus groups) were based on a topic guide designed to gather in-depth information about the practice setting and to explore sense-making and engagement (See SFile 4). These were audio-recorded. The informal interviews were short meetings with GPs about the research and its progress. These were recorded as researcher debriefing notes. 

Reviewer #1: The proposed alternative questions if Table 5 are quite interesting, because at least to me they suggest that the participants may not have understood what the HSE questions are trying to get at. The reframed questions seem to be directly either about the service experience (e.g. do you prefer a male or female gynaecologist; do you need an interpreter) or about things that more directly link to health (what foods do you eat; is there something in your travel history that could explain your symptoms) – but obviously, that is not what the HSE questions are trying to get at. If you decide to keep Table 5 and the immediately preceding paragraph, then I think that you do need to comment on this in the main text of the article, and offer a bit of a reflection about what this means for your overall results? Is this a limitation to your study perhaps?

RESPONSE: The alternative questions were proposed by participants because they thought that they were better questions to ask if the HSE was interested in adapting health services to meet their needs. It is not a limitation of the study because this is precisely the kind of data we were interested in: whether and how participants made sense of the idea of an ethnic identifier and the actual questions used to collect ethnicity data. 

This is why we say under Implications for policy and practice that our findings could be used to appraise and update HSE resources (information leaflets and training materials) to strengthen further the link between the ethnic identifier questions and health service improvement. We have edited that material to make this point clearer – see page 27.

Reviewer #2: A study that is useful in Primary Care beyond the Irish context. Could have been even better if paired with a systematic review of reasons (see Sofaer and Strech) as for example used by Moscrop et al to look at socio-economic (asking and coding) questions in the consultation. Whilst the number of GPs participating was low for this kind of study, the reasons for this make sense

RESPONSE: Thank you for positive comments about the study and GP participation. 

Thank you for the two references and we acknowledge the merits of a systematic review of reasons. Our grant, however, prioritised resources for an in-depth NPT informed analysis of three cases. As per page 6, and in line with your comment here about the usefulness of our study beyond the Irish context, these are a vehicle to learn about the collection of ethnicity data in primary care in Ireland and can be used to draw broader lessons that can be transferred to other general practice and primary care settings (conceptual generalizability).

---

## [Editor Report · Decision Letter 1]

22 Apr 2021

‘If relevant, yes; if not, no’: General practitioner (GP) users and GP perceptions about asking ethnicity questions in Irish general practice: A qualitative analysis using Normalization Process Theory.

PONE-D-20-17815R1

Dear Prof. MacFarlane,

We’re pleased to inform you that your manuscript has been judged scientifically suitable for publication and will be formally accepted for publication once it meets all outstanding technical requirements.

Kind regards,

Barbara Schouten

Academic Editor

PLOS ONE

---

## [Editor Report · Acceptance letter]

3 May 2021

PONE-D-20-17815R1 

‘If relevant, yes; if not, no’: General practitioner (GP) users and GP perceptions about asking ethnicity questions in Irish general practice: A qualitative analysis using Normalization Process Theory. 

Dear Dr. MacFarlane:

I'm pleased to inform you that your manuscript has been deemed suitable for publication in PLOS ONE. Congratulations! Your manuscript is now with our production department. 

Kind regards, 

on behalf of

Dr. Barbara Schouten 

Academic Editor

PLOS ONE